# Transfer of Self-Fruitfulness to Cultivated Almond from Peach and Wild Almond

**Thomas M. Gradziel**

Department of Plant Sciences, One Shields Avenue, University of California, Davis, CA 95616, USA;
tmgradziel@ucdavis.edu

**Abstract:** The almond [*Prunus dulcis* (Mill.) D.A. Webb] is normally self-sterile, requiring orchard placement of pollinizer cultivars and insect pollinators. Honeybees are the primary insect pollinators utilized, but climate change and the higher frequency of extreme weather events have reduced their availability to levels insufficient to meet the demands of current and anticipated almond acreage. The incorporation of self-fruitfulness may eliminate the need for both pollinizers and pollinators and allow the planting of single cultivar orchards that facilitate orchard management and reduce agrochemical inputs. Self-fruitfulness requires self-compatibility of self-pollen tube growth to fertilization, as well as a high level of consistent self-pollination or autogamy over the range of anticipated bloom environments. The Italian cultivar Tuono has been the sole source of self-compatibility for breeding programs world-wide, leading to high levels of inbreeding in current almond improvement programs. Both self-compatibility and autogamy have been successfully transferred to commercial almonds from cultivated peaches (*Prunus persica* L.), as well as wild peach and almond species. Self-compatibility was inherited as a novel major gene, but was also influenced by modifiers. Molecular markers developed for one species source often failed to function for other species' sources. Autogamy was inherited as a quantitative trait. Breeding barriers were more severe in the early stages of trait introgression, but rapidly diminished by the second to third backcross. Increasing kernel size, which was similarly inherited as a quantitative trait, was a major regulator of the introgression rate. Self-fruitfulness, along with good commercial performance of tree and nut traits, was recovered from different species sources, including *Prunus mira*, *Prunus webbii*, *P. persica*, and the *P. webbii*-derived Italian cultivar Tuono. Differences in expression of self-fruitfulness were observed, particularly during field selection at the early growth stages. Introgression of self-fruitfulness from these diverse sources also enriched overall breeding germplasm, allowing the introduction of useful traits that are not accessible within traditional germplasm.

**Keywords:** genetic diversity; inbreeding; introgression; self-compatibility; autogamy

## 1. Introduction

The almond [*Prunus dulcis* (Miller D.A. Webb] syn. *Amygdalus dulcis* Mill., *Prunus amygdalus* (L.) Batsch, *Amygdalus communis* L.) is an economically important nut-tree crop that is widely grown in Mediterranean climates. California produces over 1 billion kilograms or 85% of almonds in the global market, with a farm gate value of over USD 6 billion, making almond the state's most economically important agricultural crop [1]. The almond is normally self-sterile and commercial production requires the interplanting of cross-compatible pollinizer cultivars and the introduction during bloom of honeybee cross-pollinators that are efficient pollinators but are vulnerable to cold, windy and/or rainy conditions [2]. In addition, climate change and the higher frequency of extreme weather events have reduced the availability of commercial honeybee hives to levels insufficient to meet the demands of current and anticipated almond acreage [2]. The almond is also one of the earliest temperate tree crops to flower, typically blooming in February in California orchards, when inclement weather has historically been a major determinant of yearly variation in crop production

(Figure 1) due to insufficient cross-pollination and nut set [3]. Year-to-year California production consistency has improved in recent years, primarily due to a 3-fold increase in state-wide acreage (Figure 1), which acts to dilute regional weather-related failures that can still be devastating in affected orchards. This expanded acreage, which surpassed half a million ha in 2021 [4], has resulted in new cross-pollination challenges, including the acquisition of more than 500,000 honeybee hives for the 2 to 3-week almond bloom in order to meet the minimum recommendation of one hive per hectare for commercially effective cross-pollination [5].

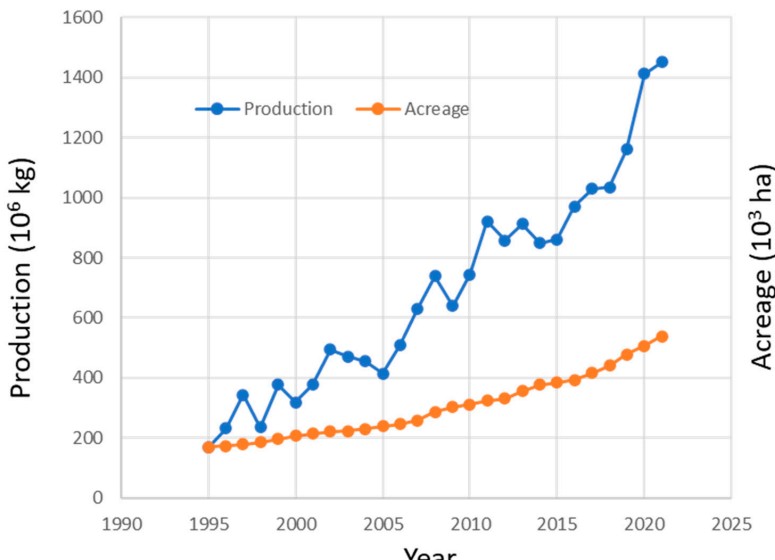

**Figure 1.** Changes in California almond kernel meat production over the past 25 years. Initial year-to-year variation is largely the consequence of weather conditions at bloom that strongly affect the amount of cross-pollination, and therefore final crop. A threefold increase in acreage, extending across the 400-mile length of the Central Valley, has diluted state-wide production variability in more recent years [1].

The introduction of self-fruitful almond selections dramatically reduces, and possibly eliminates, the dependence on and associated risks of insect pollinators. Because self-fruitful cultivars can also be planted in a solid cultivar block, they remove the need for multiple disease/insect sprays and multiple nut harvests per orchard that are required for traditional orchards where to two to five different cultivars are inter-planted by row to optimize cross-fertilization [2]. In addition to more efficient orchard management, these reductions in chemical sprays and orchard dust disturbances also result in regional improvements in air quality and associated human health.

In the major national and international breeding programs, the Italian cultivar Tuono has been the sole source of self-compatibility, leading to high levels of inbreeding [6]. In addition, the control and expression of self-compatibility is not well understood in Prunus. Early work focused on control by the major *S-locus* that contains a tightly linked S-RNase gene expressed in the pistil and the SFB gene expressed in pollen [7], although the importance of modifier genes was also noted [8]. In almond, the *S-locus* mutation that confers self-compatibility occurs in the S-RNase, while in the closely related peach, the SFB gene is affected. Defective pollen S-functions have also been reported in sweet cherry and apricot crops.

Japanese apricots with modifier genes unlinked to the *S-locus* have also been identified in these species. A double expression of the same S-genotype has been observed in Japanese plum, sweet cherry and almond crops, while control by two major loci, S and M, has recently been reported for apricots [9]. The inheritance and function have been determined for only a few of these mutations that include deletions, insertions, shift mutations and epigenetic

alterations [7]. Variable phenotypic expression of self-compatibility has also been reported in Prunus, ranging from complete self-compatibility in peaches, to variable expression in different cultivars of apricot, to almost complete expression of self-incompatibility in almond and sweet cherry crops.

Anticipating a need for expanding breeding options, a program for breeding self-fruitful almonds was established at the University of California at Davis (UCD) in the mid-1990s [10]. The goals of the program were to identify requirements for self-fruitfulness in almonds, to identify breeding sources for those traits, and to incorporate the full complement of required traits into commercially productive, California-adapted cultivars.

The almond is naturally self-sterile, a trait which encourages greater genetic diversity and environmental adaptability in seedling progeny in the often harsh environments of its central Asian origin [11]. The resulting obligate outcrossing is facilitated by the gametophytic self-incompatibility, where self-pollen is recognized and arrested within the flower pistil, preventing successful fertilization. Early research that aimed to identify cross-compatible pollinizer cultivars for commercial orchard interplanting observed inheritance patterns that were characteristic of single gene control [12].

In addition to pollen-pistil self-compatibility (SC), self-fruitfulness requires a flower structure that encourages autogamy or self-pollination. The almond produces a perfect *Prunus*-type flower (Figure 2) with an architecture very similar to that of the self-fruitful and closely related peaches. Almonds differs from peaches in that the almond style often continues to lengthen with flower age, such that the receptive stigma may be located within or well above the anthers at the time of pollen dehiscence (Figure 2). Almonds also differ from peaches in the dynamics of anther dehiscence, which is suppressed at even moderate humidity, presumably as a safeguard against pollen bursting during the frequent rain events during almond-bloom [13]. Flower development and architecture are complex and appear to be controlled by a large number of minor-affect genes [14,15].

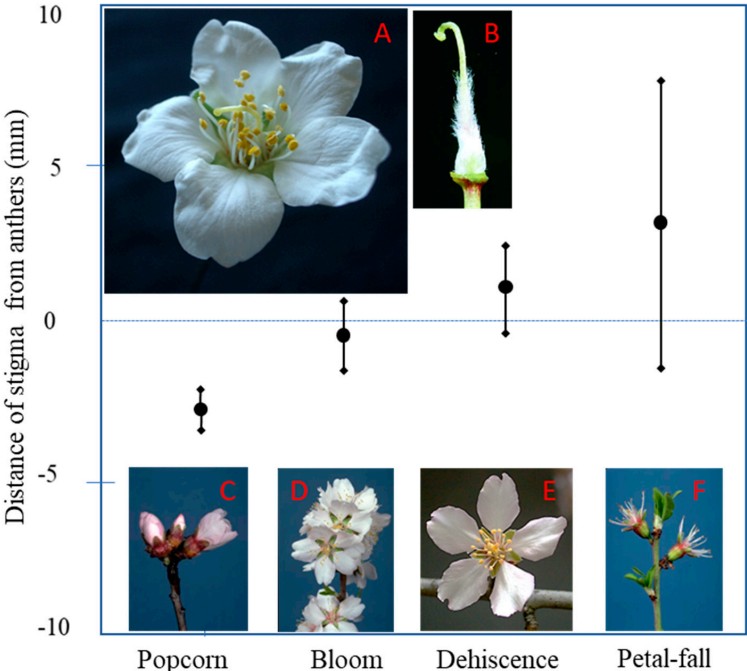

**Figure 2.** Changes in the average and standard deviation for position of the pistil stigma relative to the uppermost layer of anthers in 40 Nonpareil almond flowers at the different developmental stages. (**A**) Mature almond flower showing style-growth curving back towards dehiscing anthers. (**B**) Almond pistil at anthesis following removal of petals and floral cup showing the curved growth of the upper style. (**C**) Popcorn stage just prior to flower opening. (**D**) Flower opening or bloom before dehiscence. (**E**) Dehiscence or stage when pollen shed is first visible. (**F**) Stage between petal fall and style browning.

Consequently, the development of self-fruitfulness requires the suppression of the *S-locus* controlled recognition/arrest of self-pollen, as well as a flower structure that facilitates self-pollination or autogamy. Unlike the other stone fruit, where fruit often needs to be thinned to maximize fruit quality and market value, the almond is a relatively low-value nut crop, so that a very large proportion of viable flowers need to set to be commercially sustainable. Because almonds are grown under very diverse environments and orchard management practices ranging from dryland to highly irrigated [16], for a self-fruitful cultivar to be commercially successful, a consistently high set is required for all trees within an orchard, among differing orchard locations, and among different years, despite wide variability in environments and tree growth patterns.

This study presents tools, approaches, and techniques that have been successfully deployed to improve genetic diversity and germplasm preservation, and to facilitate full integration into almond and other nut crop breeding programs.

## 2. Materials and Methods

### 2.1. Plant Materials

Germplasm utilized in this study included cultivated almonds (P. dulcis), as well as the wild almonds *Prunus. webbii* (Spach) Vierh., *Prunus argentea* (Lam.) Rehd, *Prunus fenzliana* Fritsch and *Prunus. scoparia* (Spach), as well as cultivated peaches (*Prunus persica* (L.) Batsch) and the wild peach species (*Prunus mira* Koehne, *Prunus davidiana* (Carr.) Franch and *Prunus tangutica* (Batalin) Korsh), all of which are in the subgenus Amygdalus. When known, breeding material is identified by its cultivar name, as recorded in the work of Brooks and Olmo [17], the USDA plant introduction number, or by the UCD breeding number.

### 2.2. Controlled Crosses

All cross-pollinations to self-incompatible seed parents were carried out to flowers that had been previously enclosed in nylon mesh bags from the pink-tip to the period after the petal-fall stage to prevent unwanted insect pollinations, as detailed in the work of Kester and Gradziel [18]. All self-test-pollinations were similarly carried out to flowering shoots previously bagged to exclude insect pollinizers. Hybridizations to self-compatible parents were carried out to flowers emasculated at the popcorn-bloom stage by removing the immature floral cup and attached anthers, as described by Kester and Gradziel [18]. Pollen for hybridizations was collected and cleaned during the week before crossing and stored in a desiccator. When the pollen parent bloomed more than one week later than the seed parent, pollen was collected the previous year and stored in a desiccator at either 0 or $-40$ °C. For hybridization records, the seed parent is always presented first. A more complete description of all crossing methods is detailed by Kester and Gradziel [18].

### 2.3. Self-Compatibility Assessment

Self-compatibility was evaluated by enclosing at least 100 pre-anthesis flowers in insect-proof nylon mesh bags, followed by self-pollinating on at least three separate occasions between anthesis and initial petal fall, as described by Gradziel et al. [19]. In addition, at least 100 flowers on adjacent limbs were allowed to be open-pollinated as a flower fecundity check. Flower structures that facilitate self-pollination, particularly the pistil-stigma position relative to another position, were recorded for at least 40 flowers of selected breeding lines for changes in the position of the pistil stigma relative to the uppermost layer of anthers at the following four consecutive developmental stages: popcorn stage just prior to flower opening, flower-opening or bloom prior to dehiscence, dehiscence or the stage when pollen shed is first visible, and petal-fall or stage between petal fall and initial stigma browning. The proportion of the fruit set following controlled pollination was recorded after eight weeks from pollination for each test cross.

### 2.4. Advanced Selection Yield Evaluation

Trees were planted in 2014 on Nemaguard rootstocks in a traditional orchard design, as described by Lampinen et al. [20], at densities of 158 trees ha$^{-1}$. Rows of the cultivar Nonpareil were alternated with four randomized replicated plantings of each selection. The parameters evaluated include bloom time, hull split date, harvest date, nut quality and yield, as described by Connell et al. [3]. Yearly yield values were computed by averaging the 4 individual replications. Observations were also recorded regarding tree size, shape, structure, vigor, bearing habit, pest vulnerability and disease susceptibility to identify possible deficiencies precluding commercial deployment.

## 3. Results and Discussion

### 3.1. Self-Compatibility

Self-compatibility and self-pollination are the norm in cultivated peaches (*Prunus persica* L.), as well as their wild relatives. Several almond cultivars and a relatively small selection within the wild almond species have also shown evidence for self-compatibility.

### 3.1.1. Almond Cultivars

Unlike the more extensively studied Solanaceae, where SI (self-incompatibility) tends to be absolute, SI in almonds appears to be inconsistent [21], with a small proportion of self-fertilization normally taking place in commercially important SI almond cultivars, such as Nonpareil and Mission (Figure 3). However, the level of self-set is typically well under 10% and so not commercially significant. The efficacy of different *S*-haplotypes varies, with some *S*-haplotypes, such as $S_{14}$, being present in the California cultivars.

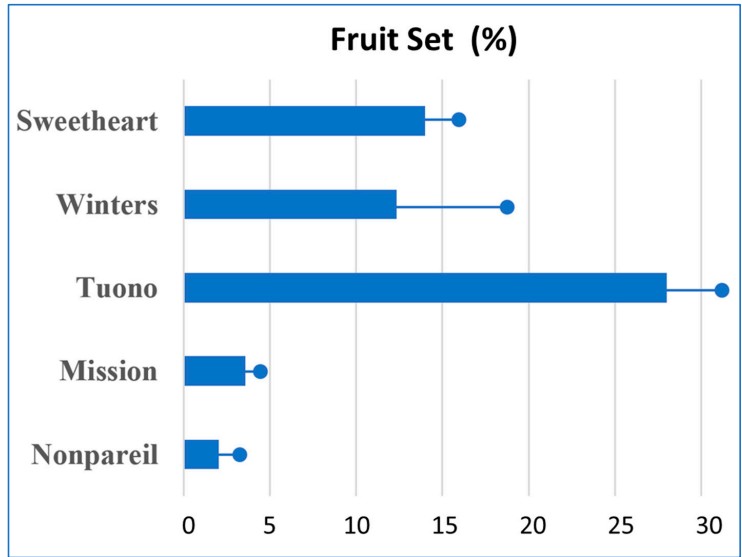

**Figure 3.** Differences in three-year average percentage fruit-set following controlled self-pollination of 100 recently opened flowers of almond cultivars classified as cross-compatible (Tuono), partially cross-compatible (Sweetheart and Winters) and self-incompatible (Mission and Nonpareil). Bars show standard deviation.

Winters [22] and Sweetheart [23] showed relatively high proportions of self-sets in some years, although not others. Although inconsistent, the $S_{14}$ allele has been incorporated in these early flowering pollinizers to improve the year-to-year cropping consistency because these early-bloom Nonpareil pollinizers typically do not have a separate source of cross-compatible pollen for their own economically important initial bloom. In addition to the *S*-locus, evidence for the presence of independent genetic modifiers of both SI and SC has been reported [24,25]. An example is the Jeffries mutation of 'Nonpareil', which results in unilateral cross-incompatibility, where cv. Jeffries can successfully pollinate 'Nonpareil',

but 'Nonpareil' pollen is incompatible on 'Jeffries' [26]. Epigenetic mechanisms that modify SI to SC have also recently been reported by Fernandez i Marti et al. [25].

In contrast, the Italian cultivar Tuono and the related cultivar Genco can be grown economically without pollinizers, as demonstrated by the analysis of self-fed pollen-tube growth [27], as well as subsequent seed set [28], under both controlled and field conditions. Numerous studies have demonstrated that SC is a result of the breakdown at the *S-locus* of pistil-based mechanisms for self-pollen recognition [5] that appears to have originated from a native self-compatible Italian tree species, *Prunus webbii*, and unconsciously introgressed to cultivated almond during the early dissemination and establishment of almond in Italy [29]. More recently developed sources for SC include the Italian cultivar Supernova [30] and the Spanish cultivar Guara [31], although molecular analysis suggests that these sources may be very similar or identical to cv. Tuono [32,33]. Because the Tuono source for SC is readily transmitted to commercially proven almond cultivars, it has been used as the primary source for developing self-fruitful cultivars by breeding programs in Spain, France, Italy, California and Australia with a consequent high level of inbreeding in recently released cultivars [6] that may be associated with inbreeding depression and declining yields [21].

### 3.1.2. Related Almond Species

Most of the almond species studied that originate from Central Europe to Western China, including *Prunus fenzliana*, which is the proposed parent species of cultivated almonds [11], are self-incompatible [34]. However, isolated accessions, such as the previously reported *P. webbii* accessions naturally found in Italy and the Balkans, have been shown to be self-compatible, presumably as a result of the mutation of the pistil-based self-recognition mechanism controlled by the *S-locus*, as first proposed by Grasselly and Olivier [27]. More recent surveys have also identified SC sources in other accessions of *P. webbii* from the Balkans, as well as isolated accessions of *P. argentea* and *P. fenzliana* [10].

### 3.1.3. Related Peach Species

Cultivated peach and its wild relatives, including *P. mira* and *P. davidiana*, are naturally self-fruitful and so self-compatible [35]. The common ancestor to both peach and almond is believed to have been self-incompatible, with self-compatibility occurring more recently in peaches [36]. Self-compatibility is controlled within the same *S-locus* as almond, although the mutation that confers SC occurs within the pollen self-recognition component [36,37] rather than the pistil-based mechanism, as presumably occurs with the almond species. Consequently, peaches serve as a unique source of SC, as well as a source that has proven itself to be effective and stable over the diverse central and east Asian environments in which peach species have evolved.

### 3.2. Autogamy

Cultivated almond and almond species typically have large, showy petals, as well as styles that often continue to elongate even after anther dehiscence (Figure 2). Peach species' flowers range from large, showy petals to the much smaller non-showy type. Unlike almond, peach styles generally show limited growth during anthesis, with the stigma position remaining within or just above the anthers at and after dehiscence. In the smaller, non-showy peach flowers, the style and receptive stigma often protrude beyond the popcorn-stage bud in a manner that would facilitate outcrossing. Peach anthers also dehisce over a much wider humidity range [13]. Anther dehiscence and self-pollination have been observed in showy peach flowers at the closed popcorn stage, suggesting that cleistogamy may be possible within this germplasm [13]. In open flowers, it is common to observe a dusting of self-pollen on the face of showy petals under windy conditions, where the petals would be blown against the dehiscing anthers. Since the receptive stigma is generally within the same region, petal flutters could also be expected to expedite self-pollination. Although outwardly very similar in appearance, peach flowers are very different from

almonds in their attractiveness to insect pollinators. While honeybees will aggressively work almond flowers for both pollen and nectar, adjacent peach flowers are largely avoided, even when the almond bloom has finished. While the cause remains unknown, it has been proposed that the honeybees avoid cyanogenic compounds present in peach pollen and nectar but largely absent in cultivated almond nectar [38].

The complexities of flower development can, thus, result in cryptic barriers to self-fruitfulness that may not be initially recognized. An example is shown in Figure 4, which shows results from initial field testing for self-fruitfulness for seedlings from advanced breeding lines derived from different SC sources. Although the presence of SC was verified with molecular markers for all selections prior to controlled self-pollination field testing, field verification of self-fruitfulness was delayed in almond and almond species sources, compared with the peach sources *P. persica* and *P. mira.* While it is normal for peach seedlings to flower and even produce fruit after 1 to 2 years of growth, almond seedlings typically do not form fruit until after 3 to 4 years of growth, despite the production of apparently normal and perfect flowers in earlier years [39]. Within the peach sources, *P. mira*-derived self-fruitful selections also appear to demonstrate greater early year-to-year consistency. Because this delayed flower fecundity occurs in the juvenile stage of the seedling trees, it represents a problem for the breeding and field-testing processes, but not commercial production, since propagated clonal plants are no longer juvenile.

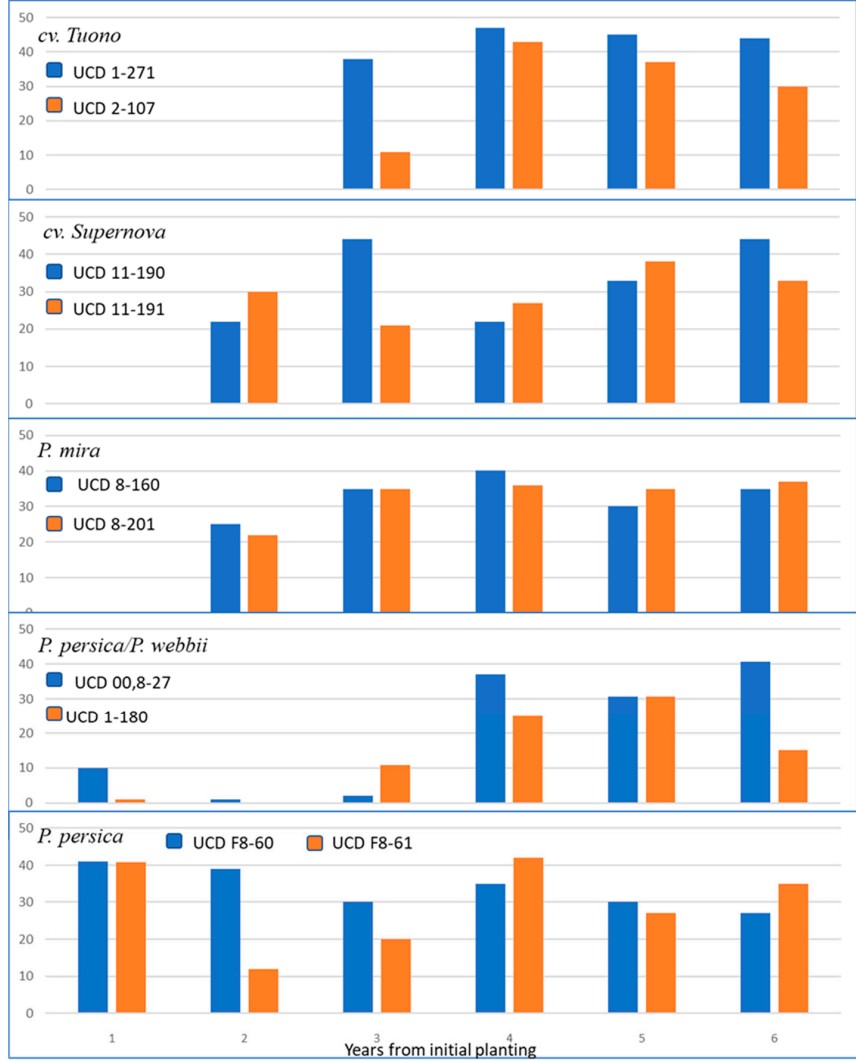

**Figure 4.** Changes in fruit-set (%) following controlled self-pollination, over six seasons of initial seedling growth for self-fruitful selections derived from different SC sources.

The complex developmental processes required for normal flower development obviate the possibility of the type of relatively direct, marker-assisted breeding possible for selecting SC [40]. Consequently, autogamy is generally considered a quantitative trait with its inherent disadvantages and advantages [21,41]. Disadvantages include the need for more complicated selection strategies, and consequently larger breeding populations, as well as the requirement for rigorous field testing to verify performance. Advantages include the opportunity to introgress a more diverse germplasm, as this would enhance breeding success not only for the targeted trait of self-fruitfulness, but would simultaneously enrich the resulting almond breeding germplasm for a range of other traits required to solve problems that arise from ongoing changes in orchard economics and regulations, as well as regional and global climates [42].

### 3.3. Breeding Strategy

The choice of breeding strategy was guided by the following three major considerations: (a) the need to capture complex traits, such as autogamy, crop productivity and kernel quality, (b) the increasing availability of effective molecular markers for identifying SI and SC, and (c) the availability at the start of the project in the mid-1990s of a diverse array of almond and peach species hybrids that were previously generated in an ongoing rootstock development program [10]. In addition to transferring self-fruitfulness to California-adapted almond cultivars, introgression of this diverse germplasm was believed to be crucial for enriching a breeding germplasm that had become critically depleted [6], as most commercial cultivars have the economically important cultivar Nonpareil as a parent, with many being the direct progeny of Mission by Nonpareil crosses [42–44]. Figure 5 summarizes the diversity of the species sources, as well as the general pattern of recurrent backcrosses for trait introgression interspersed by occasional self-pollination to facilitate the sorting out of desirable from undesirable traits, as well as ensuring that more cryptic traits that facilitate self-fruitfulness are developed. Almonds by peach hybrids typically show exceptional vigor and productivity [45] that then declines dramatically in the first 1 to 2 generations of backcrossing or self-pollination. This hybrid-breakdown [46], while occasionally resulting in novel phenotypes (Figure 6), is sorted out in subsequent breeding cycles. The transition to fruit, nut and leaf morphologies typical for almonds is rapid, often occurring within the first two to three cycles of backcrossing, as shown in Figure 7 for kernel appearance and size. Tree and flower architectures remain more variable; however, they require more focused selection. As anticipated, intraspecific breeding lines that utilize the cultivar Tuono as the source for SC showed the most rapid transition to desirable tree and kernel types, although a prevalence in this breeding lineage of undesirable small-stature trees and extensive kernel pellicle-creasing often required additional breeding cycles to correct. Despite large differences in fruit, shell and nut characteristics, breeding lines that utilize peach species as SC sources rapidly transitioned to desirable almond phenotypes [47], with a major exception being kernel mass, which responded relatively slowly to selection (see Figures 7 and 8), owing to the quantitative inheritance for this trait [39]. Unproductive (peach-like) tree architectures and kernel size below the desired mass of approximately 1g prerequisite for high commercial productivity [48] proved to be major reasons for progeny rejection after four cycles of recurrent selection. This is demonstrated in Figure 7, where selection SB6,56-88 had good kernel quality but an overly spreading tree architecture after selfing the initial peach by almond $BC_1$ selection SolSel,5-15. A second interspecific hybridization was then carried out to selection SB16, 2-44, a *P. webbii* progeny selected for self-fruitfulness, with good tree architecture and high spur density. While improving tree architecture and productivity, this sets back previous gains in kernel mass and requires two additional backcrosses to large almond-kernel seed parents before good kernel and tree qualities were fully combined in selection F8,7-179. Backcross parents were usually California-adapted almonds chosen for good kernel quality and tree architectures, as well as self-incompatibility. Self-incompatibility in the recurrent seed parent, combined with knowledge of the *S*-genotypes of the seed and pollen parents, allowed the consistent

generation of the large progeny population sizes required for combining self-fruitfulness with the multitude of quality, productivity and resistance components required for successful commercialization. For example, selection UC00,8-27 ($S_8S_F$) was from progeny resulting from its placement into a caged Nonpareil ($S_7S_8$) tree of flower bouquets of F8,7-179 ($S_7S_F$), with a commercial bumblebee hive (Koppert Biological Systems, Howell, MI USA) that was also placed in the tree cage to transfer the $S_7$ and $S_F$ donor pollen to Nonpareil flowers, where the $S_7$ would be rejected by the Nonpareil SI mechanism, ensuring that the majority of pollinations occur by the SC-conferring $S_F$ pollen. In this way, markers for S-alleles helped the breeder to pair two parents with the S7 allele to ensure that all the progeny were S7SF or S8SF [21,49]. Such an application of marker-assisted breeding precludes the need for marker-assisted selection. This is particularly valuable because molecular marker systems developed for one SC source will often not work for other distinct SC sources, thus requiring the development and co-application of effective markers for each SC source. As progeny populations approach the 10,000/year size considered desirable for making breeding progress with inherently recalcitrant tree crops [50], this type of marker-assisted selection becomes costly.

While some SI progenies are inevitably produced with this strategy because of the inconsistent nature of SI in *Prunus*, such individuals have proven useful as recurrent parents for other breeding sources in order to further enrich breeding germplasm by intermixing breeding lineages. While very efficient at generating large populations of SC progeny, this method precludes the recombination in a single individual of $S_F$ alleles from different SC sources, which might prove to be more stable under diverse environments. For example, combining $S_{f\text{-}peach}$ with $S_{f\text{-}webbii}$ from Figure 7 sources required extensive sib-and/or self-crosses, along with more tedious progeny genotyping that has slowed overall breeding progress to the point that advanced selections from such UCD lineages are only now entering the field-testing stage.

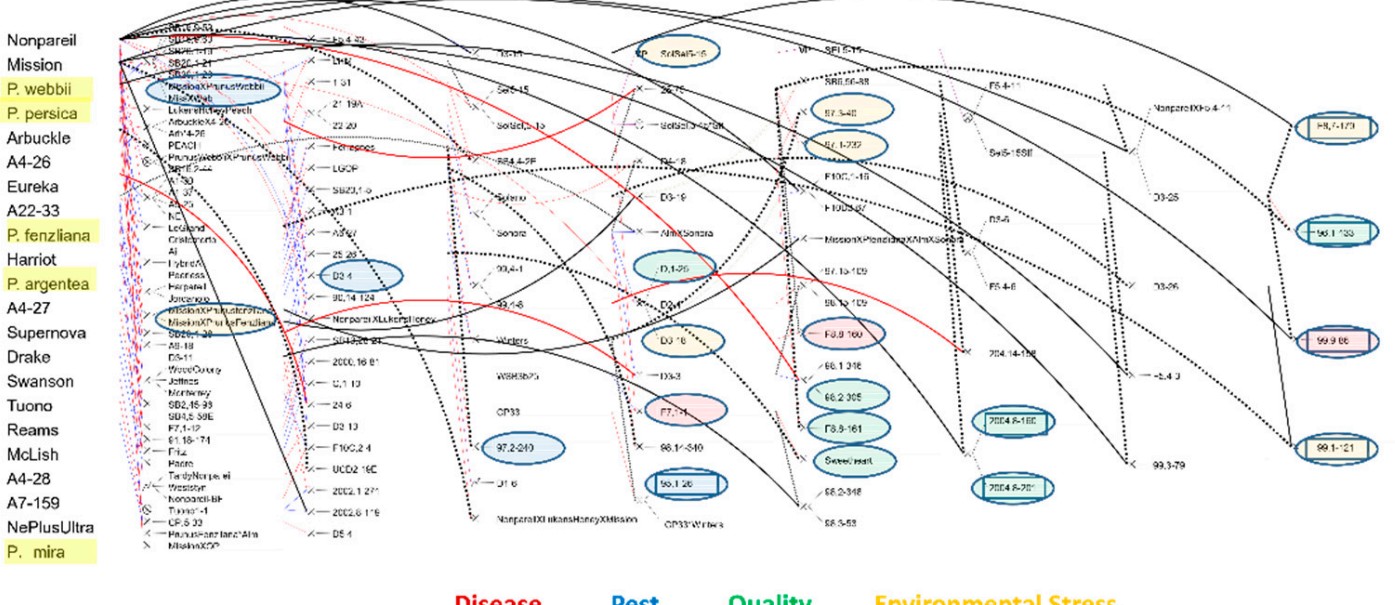

**Figure 5.** Flowchart representation of UCD breeding parents and crossing lineage for self-fruitfulness in almond. (Solid lines identify seed parents, while dotted lines identify pollen parents. Colored ovals identify germplasm showing promise as a resistant source for disease (pink), pests (blue), environmental stress (yellow) and for improved kernel quality (green) Boxed items identify promising self-fruitful advanced selections currently undergoing regional grower trials).

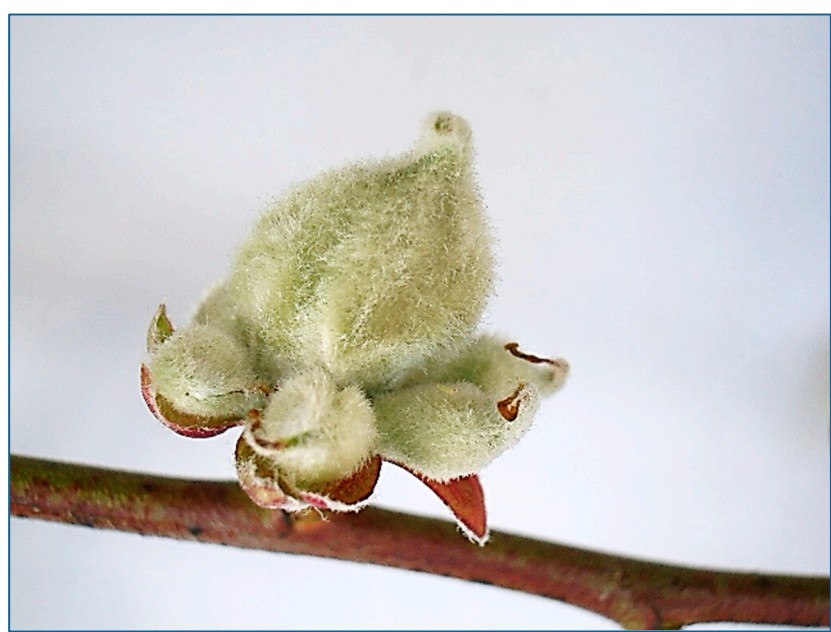

**Figure 6.** An example of the type of novel variant occasionally observed in interspecies introgression lines, showing a young fruitlet from a *P. dulcis* by *P. mira* BC1F2 where petal primordia appear to have developed as pistiloid structures.

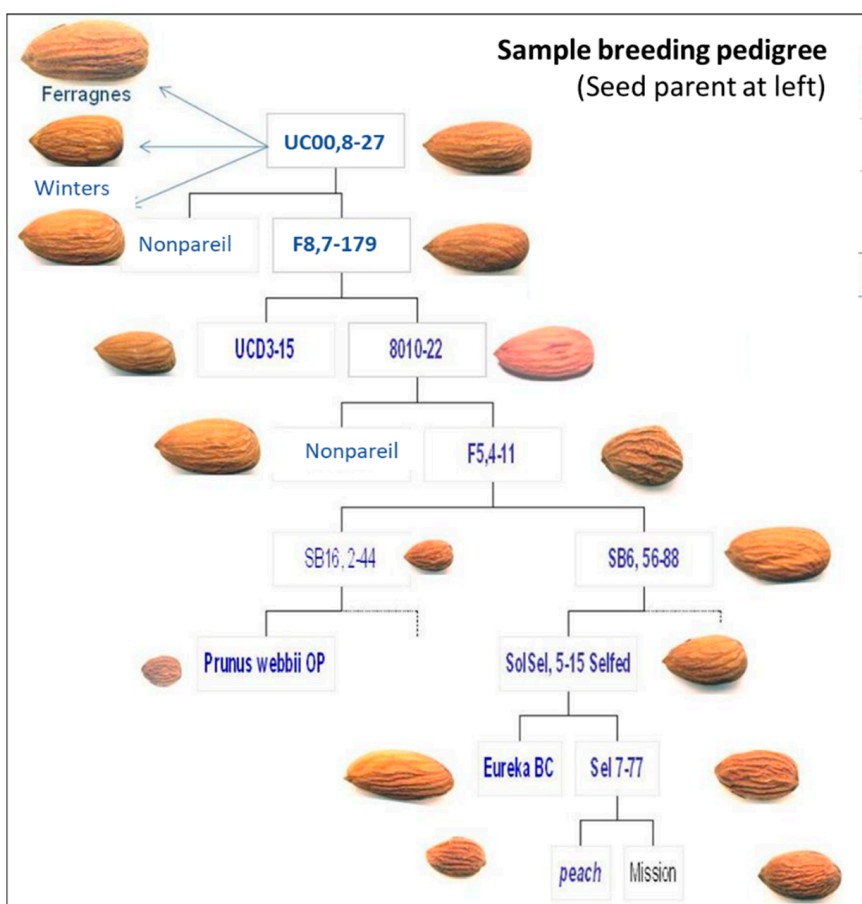

**Figure 7.** A pedigree of the UCD introgression line transferring self-fruitfulness from commercial peach, as well as *P. webbii,* seed parents. (Seed parent is on the left and pollen parent on the right).

*3.4. Field Testing*

One of the initial drivers for the development of self-fruitful almonds was to provide pollen for the early Nonpareil bloom that was crucial to high orchard yields because it had the highest proportion of viable flowers [21]. However, these early pollinizers will not have a dependable source of cross-compatible pollen for their initial and similarly economically important bloom, unless they are self-fruitful. With the recent extraordinary expansion in almond acreage, combined with increasingly regulated and costly orchard management, the options of single cultivar orchards without the need for procuring increasingly expensive honeybee hives as cross-pollinators have similarly become important market incentives for growers that choose self-fruitful cultivars. The reduction, or even elimination, in the requirement for honeybee-mediated cross-pollination should also allow more consistent year-to-year cropping because dependable self-pollination could still be achieved under inclement weather conditions that normally suppress insect-mediated cross-pollination. This is particularly relevant for the early bloom in California, as it is exposed to greater winter storm risk [3]. To evaluate overall commercial quality, including year-to-year cropping consistency, four self-fruitful advanced selections derived from 'Tuono' almond, *P. mira* and *P. persica/P. webbii* SC sources and two partially self-fruitful cultivars (Winters and Sweetheart), were compared against cv. Aldrich, which is currently the main commercial pollinizer for the early Nonpareil bloom. Trees were planted in a standardized replicated grower trial [48] located in the northern San Joaquin Valley, where inclement weather during bloom is common. The initial results, as presented in Figure 8, showed good commercial productivity for all selections while also demonstrating improved year-to-year production consistency for self-fruitful selections. The cultivar Winters, which can show relatively high though sporadic levels of self-fruitfulness because of variable expression of the $S_{14}$ haplotype [22], (see Figure 3), was the earliest of the group to flower and so was more strongly affected by the early season rainstorms in 2018. The SI cultivar Aldrich was among the last to flower, which appeared to have made it particularly vulnerable to the mid-February storms that occurred in 2019. By the end of the fifth year of production, the cultivars Aldrich, Winters and Sweetheart had the highest yields, which was in general agreement with their relatively larger tree sizes. The Tuono-derived UCD 1-271 had the smallest tree size, which, combined with its lower spur density, reduced its commercial productivity. More uniform performances were observed for the *P. mira* (UCD 8-201 and UCD 8-160] and *P. persica/P. webbii* (UCD 008-27]-derived selections, with relative productivity differences in year five that were also in general agreement with their relative tree sizes. All cultivars and selections demonstrated commercially acceptable nut and kernel quality, although the higher market classification of UCD 8-160, resulting from its desirable large-sized, oval-shaped, and blonde-pellicle-colored kernel, had made it the leader in profitability, surpassing even the more productive commercially established varieties. Because most early self-fruitful selections tend to have smaller stature trees, it was expected that they would become increasingly shaded-out by the adjacently planted Nonpareil rows in this traditionally structured orchard. However, this was only a disadvantage in traditional, multi-cultivar orchards because of the unique option of self-fruitful almonds for single-cultivar orchards, where within-row, as well as between-row, tree spacings as well as all other orchard management practices could be optimized for that cultivar.

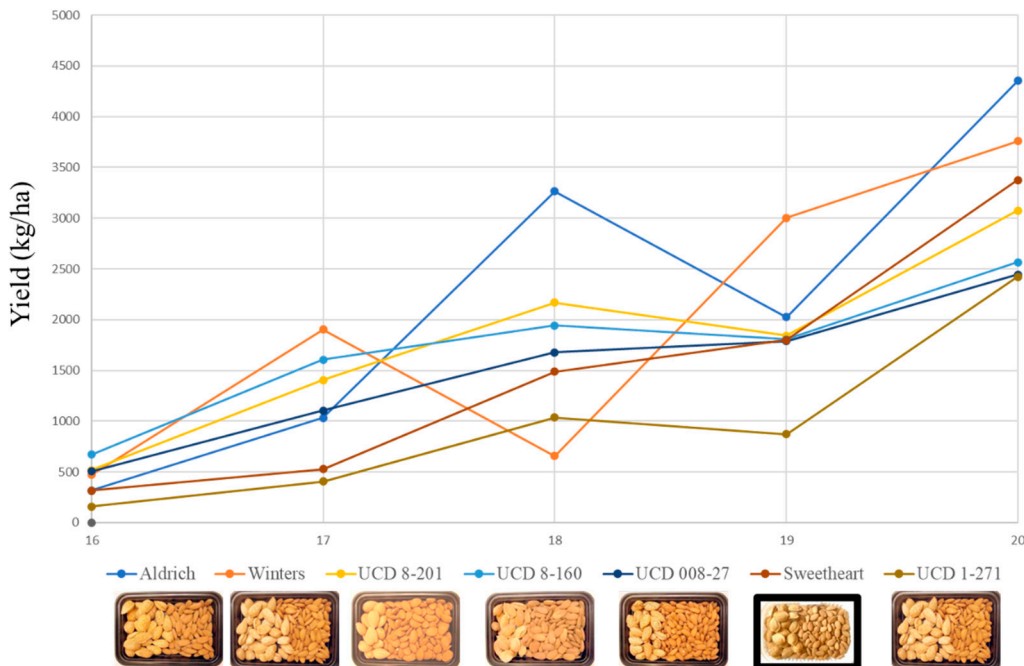

**Figure 8.** Yield progression over the first five years of commercial production for advanced UCD self-fruitful selections derived from *P. mira* (UCD 8-201, UCD 8-160), *P. persica/P. webbii* (UCD 008-27) and cv. Tuono (UCD 1-271), along with the partially self-fruitful commercial cultivars Winters and Sweetheart and the self-incompatible industry standard early-bloom pollinizer 'Aldrich'. Nut and kernel samples are from the fourth-year harvest. (Trees were planted in a traditional orchard design with cv. Nonpareil planted in alternate rows, as described by Gradziel and Lampinen [23].

## 4. Conclusions

Transforming the typically self-incompatible almond to self-fruitfulness represents a model system for the identification and integration of novel germplasm for crop improvement, in part because of the availability of effective markers for the Tuono-derived self-compatibility trait, but also because self-fruitfulness has the potential to transform an industry. Achieving the coveted goal of self-fruitful cultivars, however, requires the concurrent selection for other essential components, such as environmentally consistent autogamy. In addition, the mechanism and control of self-compatibility in Prunus remains contentious, as other forms of the SC genes have been identified but still remain largely uncharacterized [25,36,37]. Breeding commercially sustainable self-fruitful cultivars also requires the incorporation of other essential traits, such as yield, kernel and nut quality, resistance to a number of major diseases and pests, and increasingly, tolerance to progressively unstable environments, including water quantity and quality [42]. Despite the clear advantages of self-fruitfulness, previous self-fruitful cultivars developed for California in the late 1900s, including 'LeGrand', 'Garden Prince', 'All-in-One', 'Madera' and 'Self-set', have ultimately failed commercially because of deficiencies in productivity or market quality [21]. The self-fruitful cultivar Independence is currently being widely planted in the Central Valley, based on assumptions of yield and market quality comparable to cv Nonpareil, although neither assumption has thus far been validated. Self-fruitful cultivars have been traditionally planted in Italy, France and Spain, although the Italian and French industry have dramatically declined over the last 20 years, with most of the production now in Spain. Spain has twice the acreage of California but only one tenth the production [51], reflecting the traditional low-input/low-return practices. Despite extensive breeding efforts over the last few decades, the cultivar 'Guara' remains the most planted almond cultivar in Spain. This is noteworthy because recent molecular diagnostics have shown that 'Guara' is genetically identical to its presumed Tuono parent [33]. Apparently, in the early stages of field testing, the Tuono cultivar, perhaps included as the standard, became

confused with a breeding progeny. This is significant because that particular 'Tuono' tree, mistakenly identified in a multitude of breeding progeny, once again distinguished itself, demonstrating its commercial superiority. In a similar manner, the self-fruitful cultivars Mazzetto and Supernova, developed by Italian breeding programs, appear also to be misidentified clones of cv. Tuono [33]. Unlike seed propagated crops, clonally propagated crops have the potential to optimize not only genetic interactions, but also genomic and epigenetic interactions, and once optimized, as appears to be the case for commercially superior cultivars such as Tuono and Nonpareil, may be difficult to improve with the breeding tools that are largely limited to genetic manipulation, while ignoring genomic and epigenetic interactions. Ironically, the superior performance of cv. Tuono has led to decreased opportunities even for genetic manipulation. A recent survey of Spanish, French, Italian, Australian and Californian breeding programs has shown that the dependence on Tuono as the primary source for self-fruitfulness has resulted in extensive inbreeding of germplasm in these programs, with the UCD almond breeding program being the sole exception, owing to this diverse germplasm introgressed over the last few decades [6]. This enriched germplasm has also allowed the selection of traits, such as improved kernel protein and oil content, disease and insect resistance, and low nut-allergenicity, that are not available within the traditional almond breeding germplasm [47]. Ultimately, this germplasm enrichment may be of greater value in the search for sustainable solutions to the emerging almond production problems in an environment where production, market, regulatory and climate conditions will certainly change.

**Funding:** This research was funded in part by the Almond Board of California grant numbers HORT1 and HORT2.

**Informed Consent Statement:** Not applicable.

**Data Availability Statement:** Not applicable.

**Acknowledgments:** This paper is dedicated to Dale E Kester (1922–2003), Rafael Socias i Company (1946–2020) and Floyd Zaiger (1926–2020) in recognition of their pioneering research on the development of self-fruitful almond cultivars.

**Conflicts of Interest:** The author declares that the research was conducted in the absence of any commercial or financial relationships that could be construed as a potential conflict of interest.

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
