# Peer review of "Transfer of Self-Fruitfulness to Cultivated Almond from Peach and Wild Almond"

_horticulturae, doi:10.3390/horticulturae8100965_

Round 1
Reviewer 1 Report
In this manuscript, self-fruitfulness in almond is reviewed, including self-compatibility, autogamy, breeding strategies to obtain self-compatible cultivars, and the impact of this trait for both breeders and producers.
The subject is worthy of investigation and appropriate for the scope of Horticulture.
The manuscript is acceptable for publication with minor revisions. The main concern is that it is not clear whether the data shown in most figures comes from previous reports (in which case they should be properly cited in the text, figure legends, and the reference list) or whether they are new results (in which case it should be necessary to include a new Material and methods section in the manuscript).
If no new data is showed, the manuscript should be considered as a “review” instead of an “article”, following the instructions for authors guide.
If no new data is displayed, the manuscript should be considered as a "review" instead of an "article", following the author guideline.
Some specific comments:
- Self-compatibility (SC) in almond should be discussed in relation to SC in other cultivated Prunus species as apricot, sweet cherry or Japanese plum, stressing similarities and differences.
- Revise the use of “cultivar” vs “variety”, since both terms are not synonyms in the strict sense (e.g. both terms are used in the same paragraph for Tuono (L15; L26).
- Several references in the text are not included in the list of references (e.g. Fernandez i Marti et al., 2021) and vice-versa (e.g. Sorkheh et al., 2009). Some references are duplicated (e.g. References n. 21 and 22) and in other is necessary including letter (e.g. Gradziel et al., 2001)
- Figure 1. Consider changing “million” to “x 106” and “1000” to “103”
· Figure 2. If the images are not original (some intelligible text is shown in the four photos below), please include the corresponding references; if they are original, please remove the intelligible text. Include letters to identify each image (A, B,,, F), detailing each stage in the caption, especially the differences between "Bloom” and “Anthesis”. Include the reference of the data shown in the figure, or alternatively add a new Material and methods section. The same for figures 3-8.
· Figure 3. Include title and/or units (%?), and remove decimals on x-axis
- Figure 7. Include the title and units on x-axis
Author Response
Reviewer 1
Comments and Suggestions for Authors
In this manuscript, self-fruitfulness in almond is reviewed, including self-compatibility, autogamy, breeding strategies to obtain self-compatible cultivars, and the impact of this trait for both breeders and producers.
The subject is worthy of investigation and appropriate for the scope of Horticulture.
The manuscript is acceptable for publication with minor revisions. The main concern is that it is not clear whether the data shown in most figures comes from previous reports (in which case they should be properly cited in the text, figure legends, and the reference list) or whether they are new results (in which case it should be necessary to include a new Material and methods section in the manuscript).
New data is presented
Material and methods section added to the manuscript.
Some specific comments:
- Self-compatibility (SC) in almond should be discussed in relation to SC in other cultivated Prunus species as apricot, sweet cherry or Japanese plum, stressing similarities and differences.
Good idea. This discussion has been added to the introduction section.
- Revise the use of “cultivar” vs “variety”, since both terms are not synonyms in the strict sense (e.g. both terms are used in the same paragraph for Tuono (L15; L26). Done.
- Several references in the text are not included in the list of references (e.g. Fernandez i Marti et al., 2021) and vice-versa (e.g. Sorkheh et al., 2009). Some references are duplicated (e.g. References n. 21 and 22) and in other is necessary including letter (e.g. Gradziel et al., 2001). References corrected.
- Figure 1. Consider changing “million” to “x 106” and “1000” to “103”
- Figure 2. If the images are not original (some intelligible text is shown in the four photos below), please include the corresponding references; if they are original, please remove the intelligible text. Include letters to identify each image (A, B,,, F), detailing each stage in the caption, especially the differences between "Bloom” and “Anthesis”. Include the reference of the data shown in the figure, or alternatively add a new Material and methods section. The same for figures 3-8. Done.
- Figure 3. Include title and/or units (%?), and remove decimals on x-axis. Done.
- Figure 7. Include the title and units on x-axis. Done.
Reviewer 2 Report
Dear Author,
this is an extremely good and valuable paper that covers all important aspects related to self-incompatibility and the possibilities of overcrossing this issue in almond cultivars.
Due to the current structure and content, I suggest that this paper proceeds as a review paper.
References in the text should be presented according to the Journal requirements.
Author Response
Dear Author,
this is an extremely good and valuable paper that covers all important aspects related to self-incompatibility and the possibilities of overcrossing this issue in almond cultivars.
Due to the current structure and content, I suggest that this paper proceeds as a review paper.
References in the text should be presented according to the Journal requirements.
I’ve been advised to postpone the formatting of references to the Horticulturae format until after full acceptance of the paper.
Submission Date
09 September 2022
Date of this review
18 Sep 2022 08:48:20
Reviewer 3 Report
Please, see attached Word file.

Author Response
All suggestions have been incorporated. I particularly appreciated your comments and suggestions concerning the current, though rarely acknowledged, ambiguity concerning the nature of the putative SC control. You are correct that a major objective of this paper is to better document the inherent complexity in breeding this type of transformative trait into a commercially viable cultivar.
I’ve been advised to postpone the formatting of references to the Horticulturae format until after full acceptance of the paper.
Reviewer 4 Report
tHE WORK IS EXCELLENT AND THE RESULTS ARE OF GREAT INTEREST FOR THE BREEDERS AND FOR THE GROWERS WHEN THEY CHOOSE THE VARIETIES TAKING INTO ACCOUNT THEIR ORIGIN
Author Response
Thank you for your support.
Comments and Suggestions for Authors
THE WORK IS EXCELLENT AND THE RESULTS ARE OF GREAT INTEREST FOR THE BREEDERS AND FOR THE GROWERS WHEN THEY CHOOSE THE VARIETIES TAKING INTO ACCOUNT THEIR ORIGIN
Submission Date
09 September 2022
Date of this review
21 Sep 2022 00:01:27